

# MinVoellmy v1: a lightweight model for simulating rapid mass movements based on a modified Voellmy rheology

Stefan Hergarten[1]

[1]Institut für Geo- und Umweltnaturwissenschaften, Albert-Ludwigs-Universität Freiburg, Albertstr. 23B, 79104 Freiburg, Germany

**Correspondence:** Stefan Hergarten
(stefan.hergarten@geologie.uni-freiburg.de)

**Abstract.** The Voellmy rheology has been widely used for simulating snow avalanches and also for rock avalanches. Recently, a modified version of this rheology was proposed. While the conventional version of Voellmy's rheology uses the sum of Coulomb friction and a velocity-dependent friction term, the modified version assigns the two terms to different regimes of velocity. The software MinVoellmy presented here provides the first numerical implementation of the modified rheology. It consists of MATLAB and Python classes, where simplicity and parsimony were the design goals. In contrast to the majority of the models in this field, MinVoellmy uses a Cartesian coordinate system and a simple upstream scheme, which turns out to be sufficient for rheologies of the Voellmy type. Numerical tests reveal that the modified Voellmy rheology reproduces the empirical relation between runout length, height drop, and volume of large rock avalanches quite well. Furthermore, there seems to be a large potential for further research on hummocky deposit morphologies and longitudinal striations. However, the MinVoellmy software is only designed for research and teaching, but not for operational use in real-world hazard assessment.

## 1 Introduction

Modeling of rapid mass movements was pushed strongly by the ideas of Savage and Hutter (1989), who extended the shallow water equations towards granular media. The shallow water equations provide a two-dimensional, depth-averaged description of flow processes with a free surface. In their original form, the shallow water equations assume that the bed and the free fluid surface are almost horizontal. However, this is not the case for typical scenarios of granular flow. In order to overcome this limitation, the Savage-Hutter model provides an extension towards thin layers on gently curved surfaces. Beyond the general formalism, the Savage-Hutter model also includes an approximation for the stresses arising from internal deformation of a medium with a given angle of internal friction.

The idea behind the Savage-Hutter model is adopted by almost all two-dimensional continuum models of granular flow over a given topography. Existing models differ mainly concerning rheology, coordinate system and approach to reduce numerical diffusion. An overview over some of the available models is given by McDougall (2017).





In the context of snow and rock avalanches, the rheology proposed by Voellmy (1955) is widely used. It assumes a shear stress of

$$\tau = \mu\sigma + \frac{\rho g}{\xi} v^2. \tag{1}$$

at the bed. The first term describes Coulomb friction with a coefficient $\mu$ where $\sigma$ is the normal stress. The second term was adopted from the respective relation for turbulent flow of water in open channels with a rough bed, where $\rho$, $g$, and $v$ are density, gravity, and vertically averaged velocity, respectively. The parameter $\xi$ refers to the roughness of the bed. As detailed by Salm (1993), Eq. (1) does not imply turbulent flow in the sense of a complete mixing of the granular medium, which would be incompatible with the preservation of stratigraphy often found in deposits of rock avalanches (Dufresne et al., 2016). The second term in Eq. (1) can be interpreted as the result of converting kinetic energy of translation parallel to the bed into random particle motion (see also Hergarten, 2023c).

Jop et al. (2005) (see also Jop et al., 2006) derived an alternative rheology from theoretical scaling arguments in combination with laboratory experiments. This rheology was also applied to rock avalanches (Lucas et al., 2014). Similarities and differences between the two rheologies were discussed by Hergarten (2023c). Both rheologies share the property that friction increases with velocity, but effectively decreases with increasing thickness of the moving layer. The increase with velocity, however, differs strongly. While friction increases quadratically with $v$ for Voellmy's rheology, it approaches an upper limit for the rheology proposed by Jop et al. (2005). As a major limitation, both rheologies predict the lowest friction a low velocities under all conditions. As a consequence, interpreting the first term in Eq. (1) (or the respective term in the alternative rheology) as Coulomb friction is not consistent with the long runout typically observed for large rock avalanches.

Hergarten (2023c) proposed a modification of Voellmy's rheology to overcome this limitation. Instead of adding the two contributions in Eq. (1), a transition between two distinct regimes of movement in the form

$$\tau = \begin{cases} \mu\sigma & \text{for} \quad v < v_{\mathrm{c}} \\ \frac{\rho g}{\xi} v^2 & \quad v \geq v_{\mathrm{c}} \end{cases} \tag{2}$$

was assumed. While a given constant crossover velocity $v_{\mathrm{c}}$ is the simplest idea, Hergarten (2023c) also developed a model for the dependence of $v_{\mathrm{c}}$ on the thickness $h$ of the layer. This model was obtained by reinterpreting the random kinetic energy (RKE) model (Buser and Bartelt, 2009; Bartelt and Buser, 2010), which describes the supply of kinetic energy of random particle motion and its consumption. Introducing some simplifying assumptions, the relation

$$v_{\mathrm{c}} \propto \sqrt[3]{\xi h} \tag{3}$$

was obtained. This approach turned out to predict the scaling relation between volume and runout length of rock avalanches better than the version with constant $v_{\mathrm{c}}$ and is therefore used in the following.

Concerning the implementation in numerical models, numerical diffusion is typically the most serious problem. Numerical diffusion causes a progressive smoothing of sharp fronts and an artificial damping of waves. In the context of granular media, smoothing of fronts is the major problem.



Lagrangian methods are the straightforward approach to avoid numerical diffusion. In contrast to Eulerian methods, they use a coordinate system moving with the particles. In general, however, Lagrangian methods are complicated. The model DAN3D (McDougall, 2006) seems to be the only Lagrangian model in this field. It implements the concept of smooth particles, which is much simpler than a classical Lagrangian approach. In turn, the vast majority of the available models uses the Eulerian approach with a fixed coordinate system.

The total variation diminishing non-oscillatory central differencing (TVD-NOC) scheme introduced by Nessyahu and Tadmor (1990) turned out to be quite powerful in reducing numerical diffusion without introducing strong artificial oscillations. It is implemented in several models, e.g., the quite comprehensive model r.avaflow (Mergili et al., 2017). In turn, however, it will be shown in Sect. 5.2 that numerical diffusion is not a huge problem in combination with rheologies of the Voellmy type. Practically, even the simple upstream scheme works reasonable well here, which allows for simple and lightweight implementations.

The simplest form of the Savage-Hutter model refers to a channel and uses a coordinate system aligned to the bed with the $x$-coordinate in the principal flow direction. Using a curvilinear coordinate system in this spirit on an arbitrary topography is, however, not feasible. Therefore, simpler approaches are typically preferred. The model RAMMS (Christen et al., 2010) widely used in practical applications uses a coordinate system with the $x$- and $y$-coordinates aligned to the Cartesian axes, but locally inclined to become parallel to the topography. However, this coordinate system does not only involve a profile curvature along the axes, but is also non-orthogonal. The limitations arising form these properties can be overcome by introducing more or less complicated correction terms in the equations (Fischer et al., 2012).

As an alternative, some models use a Cartesian coordinate system (Bouchut and Westdickenberg, 2004; Denlinger and Iverson, 2004; Hergarten and Robl, 2015; Rauter and Tuković, 2018). Here the challenge is that the velocity at the bed must be parallel to the bed. The approach of Hergarten and Robl (2015) was even simplified in such a way that a solver for the original shallow water equations could be used. In turn, however, only the balance of the horizontal components of the momentum was considered, which introduces a serious limitation for scenarios with a strong profile curvature.

In the following, some kind of minimum implementation of the modified Voellmy rheology (Eqs. 2 and 3) will be developed. The model uses a Cartesian coordinate system in combination with an approximation to the driving acceleration by gravity. It is designed for simple applications in research, but may also be useful in teaching since the code can be fully understood with limited knowledge about numerics and is short enough to be transferred to different programming languages easily.

In turn, it is not intended to compete with comprehensive models such as r.avaflow (Mergili et al., 2017), which even includes direct coupling to a GIS and options for multi-phase flow (Pudasaini and Mergili, 2019). Even more important, it should not be used for operational hazard assessment. The model RAMMS widely used in this context has not only a much longer history of continuous development, but also includes estimates of its model parameters based on a large number of studies, which are essential for real-world applications.





## 2  Coordinates and governing equations

The model MinVoellmy presented in this paper uses Cartesian coordinates with the topography of the bed $b(x,y)$. The time-dependent model variables are the thickness of the mobile layer $h(x,y,t)$ and the velocity vector $\boldsymbol{v}(x,y,t)$. As in all models based on the theory developed by Savage and Hutter (1989), $\boldsymbol{v}$ is the component of the depth-averaged velocity parallel to the bed.

Using Cartesian coordinates circumvents several problems arising from non-orthogonal or curvilinear coordinate systems aligned to the topography. In turn, the treatment of the velocity is more complicated. The condition that $\boldsymbol{v}(x,y,t)$ must be parallel to the bed requires

$$\boldsymbol{v}\cdot\boldsymbol{n}=0 \tag{4}$$

with the normal vector

$$\boldsymbol{n}=\cos\beta\begin{pmatrix}-\nabla b\\1\end{pmatrix}, \tag{5}$$

where $\nabla b$ is the two-dimensional gradient of the bed and $\beta$ the slope angle ($\tan\beta=|\nabla b|$). While this relation would allow for reducing the velocity vector to two components in the equations, it is kept as a three-component vector here, as already proposed by Rauter and Tuković (2018). As a major difference, however, the thickness is not measured normal to the bed, but vertically. This treatment simplifies the balance of mass and momentum (Sect. 2.1), but in turn requires an approximation for the acceleration by gravity (Sect. 2.2).

### 2.1  The balance of mass and momentum

Assuming a constant density, the mass balance equation is an advection equation for the thickness $h$,

$$\frac{\partial h}{\partial t}+\frac{\partial v_{\mathrm{x}}h}{\partial x}+\frac{\partial v_{\mathrm{y}}h}{\partial y}=0. \tag{6}$$

This equation is particularly simple because $h$ is measured vertically and the horizontal movement (at velocities $v_{\mathrm{x}}$ and $v_{\mathrm{y}}$) of a vertical column is considered. The momentum balance can also be written as an advection equation in the form

$$\frac{\partial h\boldsymbol{v}}{\partial t}+\frac{\partial v_{\mathrm{x}}(h\boldsymbol{v})}{\partial x}+\frac{\partial v_{\mathrm{y}}(h\boldsymbol{v})}{\partial y}=h\left(c\boldsymbol{n}+\boldsymbol{a}-f\frac{\boldsymbol{v}}{|\boldsymbol{v}|}\right), \tag{7}$$

where $h\boldsymbol{v}$ is the depth-integrated momentum per unit mass. In contrast to Eq. (6), this equation has a nonzero source term at the right-hand side. This source term contains three contributions to the total acceleration with the first one normal to the bed and the others parallel to the bed. The first term, $c\boldsymbol{n}$, is the centripetal acceleration required to keep the velocity parallel to the bed. While other approaches (e.g., Fischer et al., 2012) compute $c$ from the local curvature, a simpler approach will be presented in Sect. 3.2. The second term, $\boldsymbol{a}$, is the gravitational acceleration parallel to the bed. The third term is the frictional deceleration in the direction opposite to the velocity, where $f$ is the absolute value.





## 2.2 The gravitational acceleration

Since the expression for $\boldsymbol{a}$ is typically derived in a bed-parallel coordinate system, its computation is briefly recapitulated in
the following. The main idea stems from the Navier-Stokes equations for an inviscid fluid, where

$$\rho\boldsymbol{a} = -\nabla p + \rho \begin{pmatrix} 0 \\ 0 \\ -g \end{pmatrix}. \tag{8}$$

The first condition to be met is $\boldsymbol{a} \cdot \boldsymbol{n} = 0$, which implies

$$\nabla p \cdot \boldsymbol{n} = -\rho g \cos \beta. \tag{9}$$

As a second condition, $p = 0$ at the free surface $s = b + h$. Then, $\nabla p$ must be normal to the surface, and thus

$$\nabla p = \lambda \begin{pmatrix} -\nabla s \\ 1 \end{pmatrix} \tag{10}$$

with an unknown factor $\lambda$. Inserting this relation into Eq. (9) yields

$$\lambda = -\frac{\rho g}{1 + \nabla s \cdot \nabla b}. \tag{11}$$

Then the pressure at the bed is

$$p_b = \nabla p \cdot \begin{pmatrix} 0 \\ 0 \\ -h \end{pmatrix} = \frac{\rho g h}{1 + \nabla s \cdot \nabla b} \tag{12}$$

and the bed-parallel acceleration

$$\boldsymbol{a} = \frac{g}{1 + \nabla s \cdot \nabla b} \begin{pmatrix} -\nabla s \\ 1 \end{pmatrix} + \begin{pmatrix} 0 \\ 0 \\ -g \end{pmatrix} = -\frac{g}{1 + \nabla s \cdot \nabla b} \begin{pmatrix} \nabla s \\ \nabla s \cdot \nabla b \end{pmatrix} = -\frac{p_b}{\rho h} \begin{pmatrix} \nabla s \\ \nabla s \cdot \nabla b \end{pmatrix}. \tag{13}$$

As a central property, the absolute value of the acceleration is

$$|\boldsymbol{a}| = g \frac{|\nabla b|}{\sqrt{1 + |\nabla b|^2}} = g \sin \beta \tag{14}$$

if the free surface is parallel to the bed ($\nabla s = \nabla b$) and zero if the free surface is horizontal ($\nabla s = \boldsymbol{0}$).

Figure 1 shows the 1-D version of Eq. (13) for slope angles $\beta = 15°$ and $\beta = 45°$, where the angle $\phi$ describes the slope of
the free surface. As a striking property, a singularity occurs at $\phi = -75°$ for $\beta = 15°$ and at $\phi = -45°$ for $\beta = 45°$. It occurs if
the denominator in Eq. (12) approaches zero, which is the case if the free surface is normal to the bed. The pressure $p_b$ grows
towards infinity then and even becomes negative after passing the singularity, which is unrealistic.

One might argue that this situation is far outside the scope of the theory proposed by Savage and Hutter (1989) and that the
occurrence of the singularity is irrelevant. Concerning numerical simulations, however, it is a big advantage if the solution is





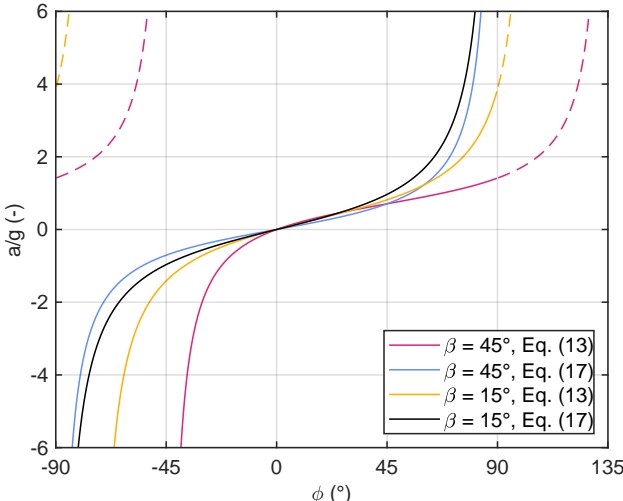

**Figure 1.** Acceleration by gravity with the original pressure at the bed (Eq. 13) and with the modified pressure (Eq. 17). The angle $\phi$ describes the slope of the free surface $s$. Negative values of $\phi$ correspond to an inclination opposite to the bed. The acceleration $a$ is $\pm|\boldsymbol{a}|$ with a positive sign downslope and a negative sign upslope.

still well-defined beyond the range of the approximations made. At this point, the widely used formulation in a local coordinate system is better than the Cartesian version. If the thickness $h$ is measured perpendicular to the bed, there is no singularity, but just $|\boldsymbol{a}| \to \infty$ for $|\nabla h| \to \infty$. The allowed range for $\phi$ in Fig. 1 would be $\phi \in (\beta - 90°, \beta + 90°)$ instead of $(-90°, 90°)$, which means that the dashed part of the curve left of the singularity would move to the right.

140     In turn, passing the singularity in the Cartesian version causes an unrealistic behavior. If an uphill-facing front ($\phi < 0°$) becomes steeper, there is an increasing outward acceleration ($a < 0$) at first. At a certain steepness, however, the acceleration changes its direction ($a > 0$), causing the material to pile up rapidly. So passing the singularity needs to be inhibited technically, e.g., by imposing a positive lower limit $d_{\min}$ to the denominator in Eq. (12),

$$p_b = \frac{\rho g h}{\max(1 + \nabla s \cdot \nabla b, d_{\min})}. \tag{15}$$

145 However, the maximum acceleration at a downhill-facing front would be limited even for $\phi \to 90°$. So there is an asymmetry in the acceleration in the form that uphill-facing fronts will typically cause a higher acceleration than downhill-facing fronts. As a consequence, a small-scale roughness of the free surface will cause an uphill acceleration in total. This issue will be investigated numerically in Sect. 5.1.

     In order to overcome this strong limitation, the model MinVoellmy uses a simplified expression for the pressure at the bed,
150 which assumes that the free surface is parallel to the bed. So $\nabla s$ is replaced by $\nabla b$ in Eq. (12), and thus

$$p_b = \frac{\rho g h}{1 + |\nabla b|^2} = \rho g h \cos^2 \beta. \tag{16}$$





This approximation can be interpreted as the hydrostatic pressure caused by the normal component of gravity, $g\cos\beta$, for a layer of a thickness of $h\cos\beta$ (perpendicular to the bed). The acceleration (Eq. 13) simplifies to

$$\boldsymbol{a} = -g\cos^2\beta \begin{pmatrix} \nabla s \\ \nabla s \cdot \nabla b \end{pmatrix} \qquad (17)$$

155   then.

This modification transfers the good properties of the original approach in a slope-aligned coordinate system to a Cartesian coordinate system. In particular, $\boldsymbol{a}$ is linear in $\nabla s$ at constant $\nabla b$. This linearity ensures that a small-scale roughness of the free surface causes no acceleration in total and that a horizontal surface ($\nabla s = \boldsymbol{0}$) is stable.

In turn, however, the curves of the two models are not tangential to each other at $\phi = \beta$, but cross each other with different slopes. Thus, Eq. (17) is not a first-order approximation to Eq. (13) concerning the difference $\phi - \beta$ for $\phi \approx \beta$. The acceleration is predicted correctly if the free surface is parallel to the bed, but the effect of an inclination of the free surface relative to the bed is overestimated. It will be shown that this overestimation has a minor effect on avalanche fronts even on steep slopes. It may, however, have a stronger effect on the tails of avalanches and on the propagation of waves at the free surface. There might be options to achieve a better approximation for $\phi \approx \beta$ preserving the good properties of the equation. For the sake of simplicity and parsimony, however, the simple approximation is used in the following.

### 2.3   Friction

The friction term in Eq. (7) refers to the acceleration of a vertical column. Here it has to be taken into account that the shear stress $\tau$ at the bed acts on the inclined bed area, which is locally by a factor of $\frac{1}{\cos\beta}$ greater than the horizontal area. Then the deceleration is

$$f = \frac{\tau}{\rho h \cos\beta} \qquad (18)$$

with the shear stress $\tau$ from Eq. (2). For $v \geq v_c$, we obtain

$$f = \frac{g}{\xi h \cos\beta} |\boldsymbol{v}|^2 \qquad (19)$$

without further assumptions. In the range of Coulomb friction ($v < v_c$), however, the normal stress at the bed must be specified. For simplicity, effects of internal deformation of the granular medium (typically expressed in terms of the so-called earth pressure coefficients) are neglected in the following. If the friction arising from the centripetal acceleration is also also neglected, the normal stress $\sigma$ is given by the pressure at the bed $p_b$, which leads to

$$f = \mu \frac{p_b}{\rho h \cos\beta}. \qquad (20)$$

The increase in friction due to the centripetal acceleration can easily be included in the form

$$f = \mu \left( \frac{p_b}{\rho h \cos\beta} + c \right) = \mu \left( g\cos\beta + c \right) \qquad (21)$$





in combination with the modified pressure at the bed (Eq. 16). On concave profiles, however, $f$ may even become negative. In this situation, the mobile material would detach from the bed, which is not captured by models of this type. Since an acceleration by friction is unrealistic, negative values of $f$ should be replaced by 0, and thus

$$f = \begin{cases} \mu \min\left(g\cos\beta + c, 0\right) & v < v_{\mathrm{c}} \\ \frac{g}{\xi h \cos\beta}|\boldsymbol{v}|^2 & v \geq v_{\mathrm{c}} \end{cases} \quad \text{for} \quad . \tag{22}$$

## 3  Numerical implementation

The numerical implementation uses a regular grid with a constant spacing $\delta x$ in the $x$-direction and $\delta y$ in the $y$-direction. The variables are the thickness $h$ and the depth-integrated momentum per unit mass $h\boldsymbol{v}$ as a three-component vector. Both variables are considered at the nodes $(x, y)$.

### 3.1  Mass balance

The mass balance (Eq. 6) is an advection equation without source terms. An explicit Euler scheme is used in combination with
an upstream discretization. The question why this simple scheme is sufficient for the Voellmy rheology considered here will be addressed in Sect. 5.2. Applying the explicit Euler scheme to Eq. (6) yields

$$h(t + \delta t) = h(t) - \delta t\left(\frac{\partial v_{\mathrm{x}}h(t)}{\partial x} + \frac{\partial v_{\mathrm{y}}h(t)}{\partial y}\right) \tag{23}$$

First, the values $v_{\mathrm{x}}$ and $v_{\mathrm{y}}$ are computed at the nodes from $h\boldsymbol{v}$ and $h$. Then the values of $v_{\mathrm{x}}$ are interpolated linearly to the points $(x \pm \frac{\delta x}{2}, y)$. Depending on the sign of this velocity, $h$ at either of the nodes (the upstream point) is adopted to $(x \pm \frac{\delta x}{2}, y)$, and
a central difference quotient is used for the $x$-derivative at $(x, y)$. The same procedure is applied to the $y$-derivative.

### 3.2  Momentum balance

The momentum balance (Eq. 7) is separated into several steps.

**Step 1: advection**

The first step is basically the same as for $h$, except that the result is not $(h\boldsymbol{v})(t + \delta t)$, but an intermediate value

$$(h\boldsymbol{v})' = (h\boldsymbol{v})(t) - \delta t\left(\frac{\partial v_{\mathrm{x}}(h\boldsymbol{v})(t)}{\partial x} + \frac{\partial v_{\mathrm{y}}(h\boldsymbol{v})(t)}{\partial y}\right). \tag{24}$$

**Step 2: centripetal acceleration**

The second step addresses the first term at the right-hand side of Eq. (7) and computes a second intermediate value

$$(h\boldsymbol{v})'' = (h\boldsymbol{v})' + \delta t\, h c \boldsymbol{n}. \tag{25}$$





The centripetal acceleration $c$ is obtained from the condition that $\boldsymbol{v}$ must be parallel to the bed ($(h\boldsymbol{v})'' \cdot \boldsymbol{n} = 0$), which yields

$$\delta t\, h c = -(h\boldsymbol{v})' \cdot \boldsymbol{n}. \tag{26}$$

Since the centripetal acceleration should not change the absolute value of the velocity, $(h\boldsymbol{v})''$ is then rescaled in such a way that $|(h\boldsymbol{v})''| = |(h\boldsymbol{v})'|$.

**Step 3: gravitational acceleration**

This step generates the next intermediate values

$$(h\boldsymbol{v})''' = (h\boldsymbol{v})'' + \delta t\, h \boldsymbol{a}. \tag{27}$$

The gravitational acceleration $\boldsymbol{a}$ parallel to the bed (Eq. 13 or 17) requires the gradient of the free surface $s = b + h$. In order to avoid checkerboard problems, one-sided difference quotients are used, although central difference quotients would yield a better accuracy theoretically. If central difference quotients were used, a checkerboard pattern with one value of $s$ at the black fields of a checkerboard and another value at the red fields would yield no acceleration and thus be stable. One-sided difference

quotients in the direction of steepest descent (or smallest increase) in $s$ turned out to be most robust.

**Step 4: friction**

The final step has the form

$$(h\boldsymbol{v})(t + \delta t) = (h\boldsymbol{v})''' - \delta t\, h f \frac{\boldsymbol{v}}{|\boldsymbol{v}|}. \tag{28}$$

Since friction is opposite to the velocity, $(h\boldsymbol{v})(t + \delta t)$ is obtained by rescaling $(h\boldsymbol{v})'''$.

For $v < v_c$, the amount in $hv$ consumed by friction is $\delta t\, h \mu \min(g\cos\beta + c, 0)$ according to Eq. (22). If this amount exceeds the actual amount $|(h\boldsymbol{v})'''|$, it is assumed that the movement stops. Accordingly, the length of the vector $(h\boldsymbol{v})(t + \delta t)$ is

$$|(h\boldsymbol{v})(t + \delta t)| = \max\left(|(h\boldsymbol{v})'''| - \delta t\, h \mu \min(g\cos\beta + c, 0), 0\right), \tag{29}$$

where the outer maximum function takes the stopping criterion into account.

For $v \geq v_c$, we have to take into account that friction depends on velocity. Since $f \to \infty$ for $h \to 0$ according to Eq. (22), an

implicit scheme is used here in order to avoid an additional limitation to $\delta t$. This means that $\boldsymbol{v}$ in Eq. (22) is expressed in terms of $(h\boldsymbol{v})(t + \delta t)$, which yields

$$|(h\boldsymbol{v})(t + \delta t)| = |(h\boldsymbol{v})'''| - \frac{g\delta t}{\xi h^2 \cos\beta}|(h\boldsymbol{v})(t + \delta t)|^2. \tag{30}$$

This is a quadratic equation in $|(h\boldsymbol{v})(t + \delta t)|$. It is solved by

$$|(h\boldsymbol{v})(t + \delta t)| = \sqrt{\gamma^2 + 2\gamma|(h\boldsymbol{v})'''|} - \gamma \tag{31}$$

with

$$\gamma = \frac{\xi h^2 \cos\beta}{2g\delta t}. \tag{32}$$





## 4 Software description

At present, MATLAB and Python implementations of MinVoellmy are available under the GNU General Public License. None of them requires specific packages, except for NumPy for the Python version. Each version contains separate classes for the one- and two dimensional versions. The implementation is minimalistic. The classes contain a constructor and a method `step` for performing a forward time step, but neither methods for input/output nor graphics components.

The constructor requires six (1D) or seven (2D) mandatory arguments:

– Two arrays `b` and `h` for the bedrock elevation $b$ and the initial thickness $h$.

– The grid spacing `dx` and `dy` (only for the 2D version).

– The physical parameters $\mu$ (`mu`), $\xi$ (`xi`), and $v_c$ (`vc`). The latter describes the crossover velocity at a thickness $h = 1$ m, and the actual value of $v_c$ is computed from Eq. (3). Values $v_c \leq 0$ switch to the conventional Voellmy rheology. These three parameters may be either scalar values or arrays.

Further optional arguments are:

– A minimum thickness $h_{\min}$ (`hmin`). It is assumed that material can move only if $h > h_{\min}$ (default = 0).

– The minimum value $d_{\min}$ (`dmin`) for the denominator in Eq. (15) in case the original expression for the pressure is used. The simplified expression (Eq. 16) is used for $d_{\min} = 0$, which is strongly recommended (default = 0).

– A logical value `cent` to define whether the effect of the centripetal acceleration on Coulomb friction is taken into account.

– The gravitational acceleration $g$ (`g`) (default = 9.81).

The method `step` for the forward time step updates the thickness `h` and the Cartesian components `uh`, `vh` (only in the 2D version), and `wh` of the momentum vector $h\boldsymbol{v}$. The time increment $\delta t$ (`dt`) is the only mandatory parameter. Optionally, an upper limit `cfl` for the Courant number

$$C = \frac{|v_{\mathrm{x}}|\delta t}{\delta x} + \frac{|v_{\mathrm{y}}|\delta t}{\delta y} \tag{33}$$

can be defined. In this case, $\delta t$ is reduced automatically if $C > $ `cfl`, and the reduced value of $\delta t$ is returned. This option can be used for adjusting $\delta t$ dynamically to the velocity. According to the Courant-Friedrichs-Lewy (CFL) criterion, $C > 1$ makes the explicit scheme unstable. Setting `cfl` to a sufficiently small value, e.g., 0.5, avoids instability of the advection terms. However, the CFL criterion does not capture the acceleration and friction terms, so that $\delta t$ must not be too large even when using the optional argument `cfl`. In particular, the transition in friction at $v_c$ may require quite small time increments $\delta t$.

In order to increase efficiency, the 2D version restricts the computation to a rectangle around the active region ($h > h_{\min}$) in each time step.





## 5 Numerical tests

In this section, several tests addressing the fundamental properties of the modified rheology and the numerical approach are presented. Unless stated explicity, the parameter values $\mu = 0.75$, $\xi = 250\,\mathrm{ms}^{-2}$, and $v_\mathrm{c} = 5\,\mathrm{ms}^{-1}$ (at a thickness of $h = 1$ m) are used.

### 5.1 The modified pressure

The first numerical test refers to the modification applied to the pressure at the bed in Sect. 2.2. The scenario is a pile of 100 m width and 50 m height placed on a slope with $\beta = 30°$. In order to point out the differences between the versions more clearly, it is assumed that the material is fluidized from the beginning ($\mu = 0$, $v_\mathrm{c} = 0$). The simulations are performed on a rather fine grid with $\delta x = 1$ m and a very small time increment $\delta t = 10^{-3}$ s.

A simulation with a coordinate system aligned to the bed is used as a reference. Technically, $g$ in the acceleration term has to be multiplied by $\cos\beta$ here, and $\nabla b$ has to be set to zero. In turn, an additional downslope acceleration $g\sin\beta$ has to be taken into account.

    As shown in Fig. 2, the different simulations start similarly with some waves downslope of the crest, owing to the opposite directions of the acceleration at the crest. For the scenarios with the original pressure and small lower limits $d_{\min}$ of the
denominator in Eq. (15), these waves propagate uphill rapidly. The pile even moves uphill in total for $d_{\min} \leq 0.1$, which is completely unrealistic. This artifact was already discussed in Sect. 2.2.

    In turn, the modified pressure (Eq. 16) overestimates the acceleration at the steep downslope front. As a consequence, spreading of the downslope front is stronger than in the reference scenario. The overall shape is, however, quite similar for $t \geq 4$ s. Due to the faster spreading, the layer is slightly thinner with the modified pressure, resulting in a lower velocity. So the
lead over the reference scenario melts down through time.

    The scenario with a strong limitation of the denominator ($d_{\min} = 1$) also stays close to the reference scenario for some time. However, the tail persists, while the front is too slow. These results confirm the theoretical arguments given in Sect. 2.2 that the asymmetry in the acceleration term causes artifacts. The version with the original pressure appears to be unsuitable even with the strong limitation of the denominator. In turn, the version with the modified pressure works well, except for the spreading
of the downslope front being too fast in the beginning.

### 5.2 Numerical diffusion

Numerical diffusion is typically the most challenging problem in computational fluid dynamics. In order to investigate the effect of numerical diffusion, the movement of a body of granular material moving down a straight slope with a given slope angle $\beta$ is considered. Let us focus on steady-state solutions in the sense that the entire body moves at a constant velocity
$v_0 \geq v_\mathrm{c}$ without changing its shape. Then the driving acceleration $|\boldsymbol{a}|$ must be balanced by friction at each point, so

$$|\boldsymbol{a}| = f = \frac{g}{\xi h \cos\beta} v_0^2 \qquad (34)$$



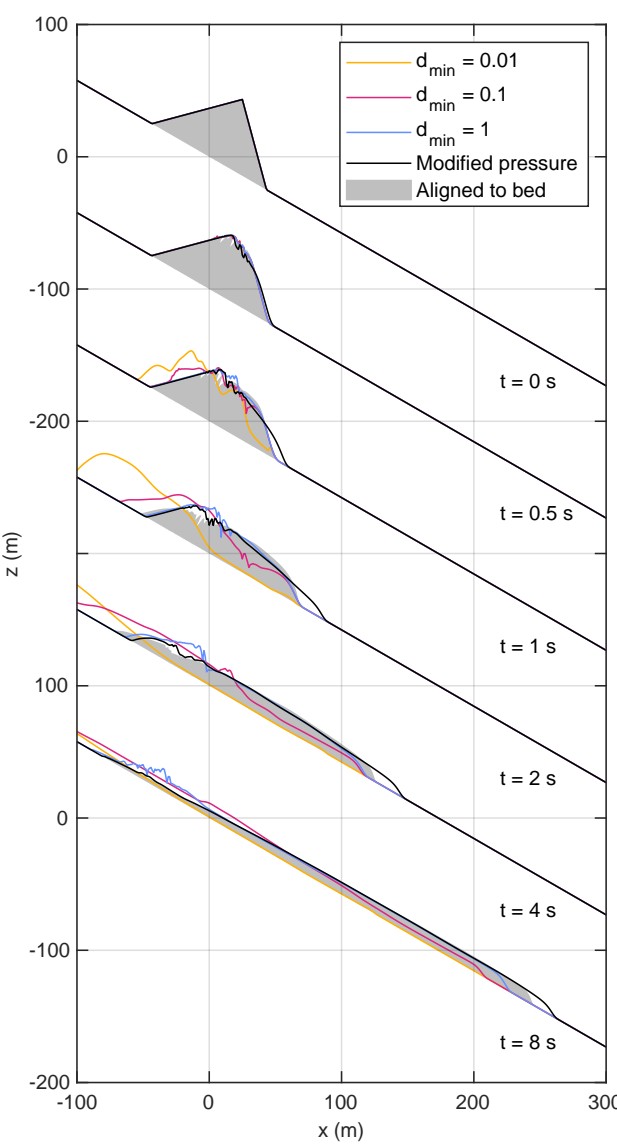

**Figure 2.** Collapse of a granular pile on a slope with $\beta = 30°$. The colored curves refer to the original pressure with different lower limits (Eq. 15), and the black line to the modified pressure (Eq. 16). The gray-shaded area shows the result of a simulation with a coordinate system aligned to the slope as a reference. Curves are shifted vertically, and labels at the $z$-axis refer to the first and last plot.





according to Eq. (19). The simplest solution is an infinite layer with a constant thickness $h_0$. Then the surface is parallel to the bed, and the acceleration is $|\boldsymbol{a}| = g\sin\beta$ (Eq. 14) for both pressure models, so that Eq. (34) yields

$$h_0 = \frac{v_0^2}{\xi \cos\beta \sin\beta}. \tag{35}$$

Although this solution does not describe a front moving downslope, it helps to write Eq. (34) in a more convenient form,

$$|\boldsymbol{a}| = g\sin\beta \frac{h_0}{h}. \tag{36}$$

In combination with the modified pressure (Eq. 17), the acceleration is

$$|\boldsymbol{a}| = g\cos\beta \left(-\frac{\partial s}{\partial x}\right) = g\cos\beta \left(\tan\beta - \frac{\partial h}{\partial x}\right), \tag{37}$$

which leads to the differential equation

$$\frac{\partial h}{\partial x} = \tan\beta \left(1 - \frac{h_0}{h}\right). \tag{38}$$

This equation can be solved analytically in the form $x(h)$ instead of $h(x)$. It is recognized by computing the derivative that the solution is

$$x = \frac{h + h_0 \ln\left(1 - \frac{h}{h_0}\right)}{\tan\beta} + \text{const.} \tag{39}$$

This analytical solution is the reference for the numerical test. It describes a front that becomes vertical ($\frac{\partial h}{\partial x} \to \infty$) for $h \to 0$,

while $h \to h_0$ for $x \to -\infty$. It is plotted as a blue line without markers in Fig. 3 for $h_0 = 10$ m.

For the numerical representation, it is assumed that $h = h_0$ and $|h\boldsymbol{v}| = h_0 v_0$ at the left-hand boundary. This leads to a front propagating with the velocity $v_0$ (parallel to the bed). Propagation is simulated over a horizontal distance of 10 km in order to approach the steady-state shape for $\delta x = 1$ m and $\delta x = 10$ m. In order to minimize the effect of $\delta t$, the small value $\delta t = 10^{-3}$ s is still used here.

As expected, the deviation from the analytical solution increases with increasing grid spacing. The artificial widening of the front is, however, limited to a few times $\delta x$. More important, this widening is limited in time. An initially sharp front is widened rapidly, but width and shape stabilize soon.

This behavior arises from a specific property of Voellmy-type rheologies. The frictional acceleration (Eq. 19) increases with decreasing thickness. So material running ahead of the front is decelerated and thus overrun be the front. As a consequence, the

shape of the front predicted by Eq. (39) is stable. The simple upstream scheme introduces numerical diffusion and widens the front, but the stability of the front counteracts this process. This is the reason why numerical diffusion is not a serious problem in combination with Voellmy-type rheologies, in contrast to many other applications of the shallow water equations.

For completeness, Fig. 3 also shows the results obtained for the original pressure (Eq. 13) and for the model with a coordinate system parallel to the bed. The analytical solution is the same for both versions, but the respective front is steeper than for the

version with the modified pressure. The numerical solution with the coordinate system aligned to the bed approximates the

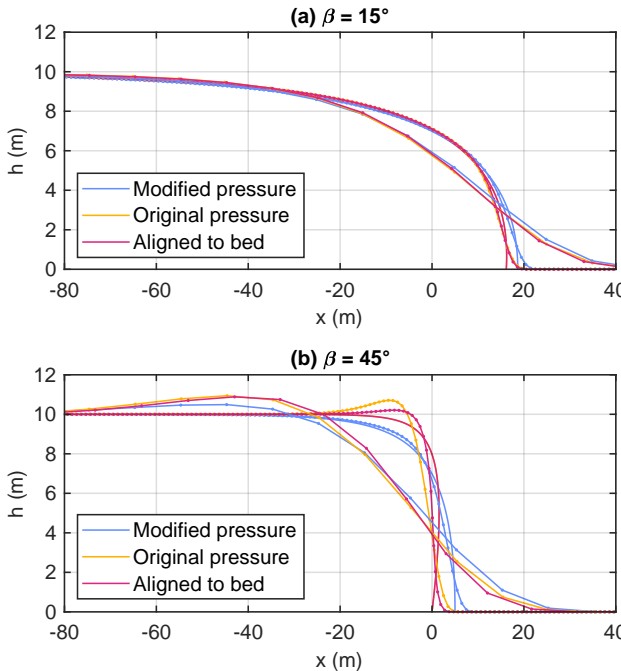

**Figure 3.** Front of a layer with a vertical thickness $h_0 = 10$ m after traveling a horizontal distance of 10 km. Curves without markers refer to the analytical solutions. Markers refer to the nodes of the numerical solutions with $\delta x = 1$ m and $\delta x = 10$ m. All curves are centered horizontally in such a way that the mean thickness over an interval from $-250$ m to 250 m is $\frac{h_0}{2}$.

front better than in Cartesian coordinates, in particular if the slope is steep. This result is not surprising since the front is normal to the bed at the bottom, while the Cartesian version is limited by a vertical front.

Overall, however, the differences between all versions are rather small. In combination with the findings of the previous section, this finding justifies the major approximations introduced in the model MinVoellmy. First, the modified pressure
provides a reasonable approximation and makes a treatment in Cartesian coordinates with the thickness measured vertically feasible. Second, the simple upstream scheme for the advection terms works reasonably well in combination with the Voellmy rheology.

### 5.3 Comparison to the conventional Voellmy rheology

Figure 4 shows the results for a slope with $\beta = 45°$ combined with a horizontal plane. The source area is a segment of an
ellipse with an aspect ratio of 4:1 and a vertical wall at the upper edge at a height of 1900 m above the runout plane. Grid spacing is $\delta x = 10$ m and time increment $\delta t = 0.01$ s. This scenario is chosen to illustrate the robustness of the approach. In combination with the modified pressure, which is also used in the following examples, the Cartesian approach works well technically, although the applicability of the Savage-Hutter theory to the thick detached body is not given here, regardless of the rheology and the numerical scheme.





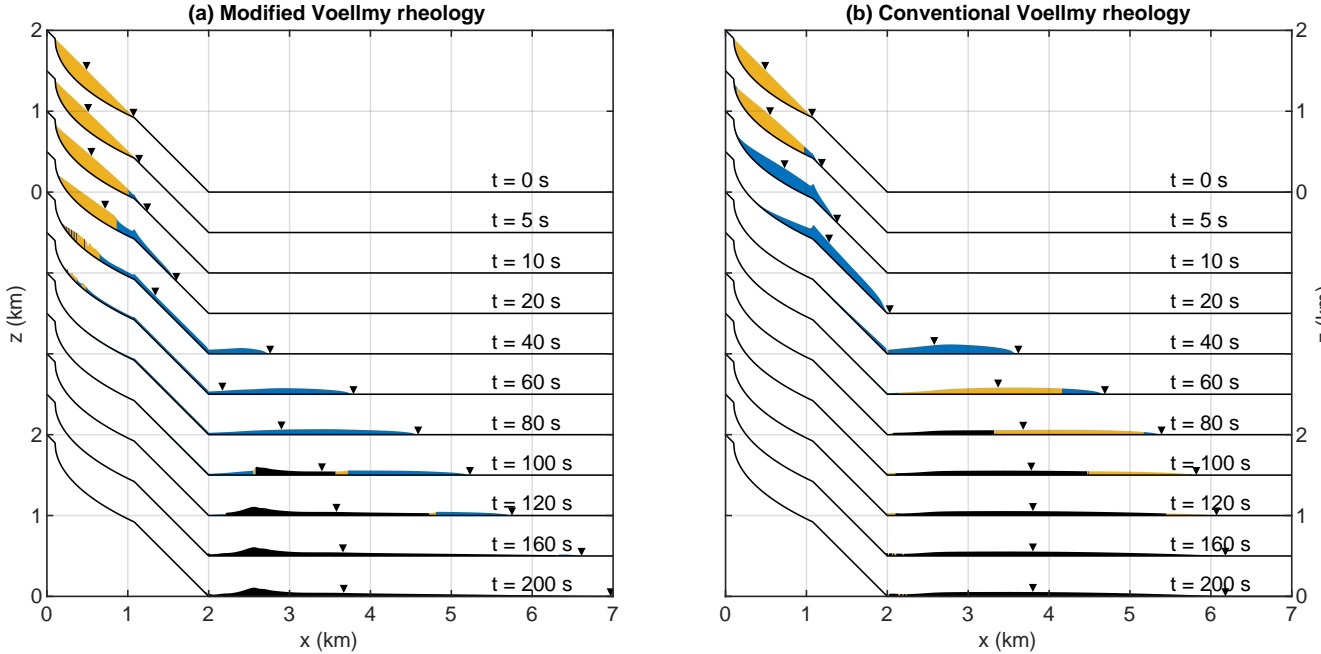

**Figure 4.** Snapshots of a 1D simulation. Orange-colored areas indicate $v \geq v_c$ for the modified Voellmy rheology (a) and that the $v^2$ friction term is stronger than the Coulomb friction term for the conventional rheology (b). Blue areas correspond to the opposite situation (Coulomb friction dominates). Black areas are at rest. The triangles depict the center of mass and the front. Curves are shifted vertically, and labels at the $z$-axis refer to the first and last plot.

For the scenario with the conventional Voellmy rheology, the Coulomb friction term has to be reduced by a factor of 10 (so $\mu = 0.075$) in order to achieve a similar runout length. The two versions differ strongly already during the phase of mobilization. Owing to the artificially reduced coefficient of friction $\mu$, the body is mobilized much faster in the conventional scenario than with the modified rheology. In order to avoid the fast mobilization, Aaron and Hungr (2016) extended the model DAN3D by considering a more or less rigid block during the first phase of movement. This might not be necessary necessary for the modified rheology. However, it should be kept in mind that we are outside the range of applicability of the Savage-Hutter theory here. So the mobilization seems to be more realistic with the modified rheology, but is not necessarily physically correct.

The thickness overshoots at the lower edge of the detachment area. This overshooting is inevitable at a sharp kink in topography. Since the absolute value of the velocity remains constant at the kink, the horizontal component decreases. Conservation of mass requires an increase in vertical thickness $h$, which causes the overshooting. This effect is, however, neither unique to the approach used here nor a serious problem.

Further differences between the two versions occur during the runout in the horizontal plane. For the modified rheology, friction increases instantaneously when the velocity drops below $v_c$, and movement stops quite soon. As a consequence, a quite large part of the mass comes to rest within a narrow time span between $t = 80$ s and $t = 100$ s. This region expands towards





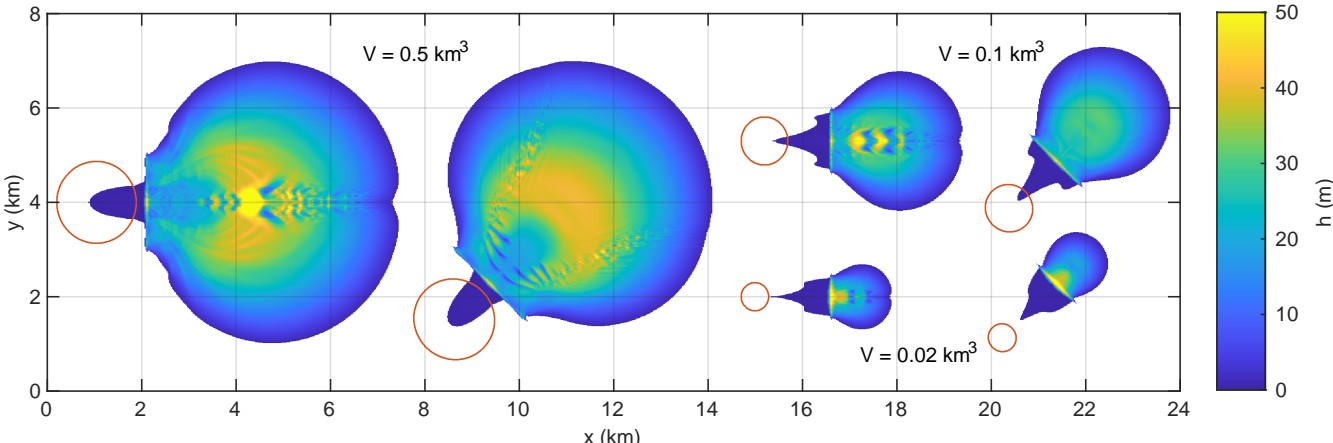

**Figure 5.** Deposits obtained from volumes of $V = 0.5$ km$^3$, $V = 0.1$ km$^3$, and $V = 0.02$ km$^3$ with axis-parallel and diagonal orientations. Only deposits with a thickness $h \geq 0.1$ m are shown. The red lines depict the detachment area at the slope.

both sides and is bounded by two narrow ranges with Coulomb friction. The material upstream of this region is piles up, which
results in a hill in the deposits.

In turn, there is a rather thin layer propagating with little friction, which finally dominates the runout length. Despite the difference in $\mu$, the maximum runout length is even bigger than for the scenario with the conventional Voellmy rheology, while the distance traveled by the center of mass is similar. This result confirms the findings of Hergarten (2023c) that the modified Voellmy rheology allows for a long runout without assuming an artificially low coefficient of friction $\mu$.

## 5.4 Two-dimensional simulations

This section mainly addresses the effect of the orientation of the grid in 2D. Starting point is the situation considered in the previous section, but extended into the $y$-direction by an ellipsoidal volume with an aspect ratio of 4:1:1. For this scenario, the total detached volume is $V = 0.1$ km$^3$. In addition, a smaller volume of $V = 0.02$ km$^3$ and a larger volume of $V = 0.5$ km$^3$ are considered, all with the same aspect ratio and the same height of 1900 m above the horizontal plane. Grid spacing is $\delta x = \delta y = 10$ m. The time increment is $\delta t = 0.1$ s with an additional upper limit $C \leq 0.25$ for the Courant number (Eq. 33). All simulations were run over a total time span of 500 s.

Figure 5 shows the final deposits of the three volumes for a slope aligned to the coordinate axes and for a slope aligned to the diagonal line in the $x$-$y$ plane. The small-scale topography of the deposits depends on the orientation of the slope. In particular, striations occur if the velocity is parallel to one of the coordinate axes. Since the principal flow direction is parallel to the $x$-axis for the axis-parallel scenario, this effect is stronger here than for the diagonal setup.

As already recognized in Sect. 5.3, the formation of hummocky deposit morphologies arises from the discontinuity in friction at $v_c$. This discontinuity also allows for the formation of striations since the governing equations do not include transverse diffusion of momentum. Owing to the advective characteristics of Eq. (7), different flow lines are in principle decoupled. So





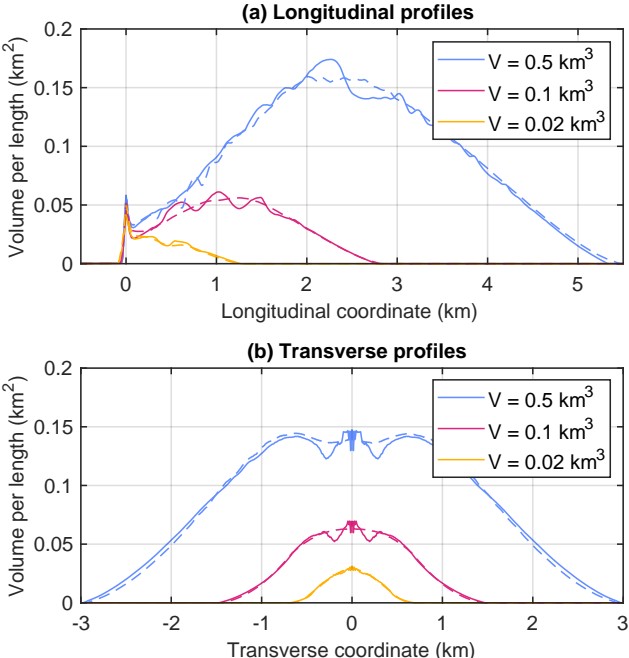

**Figure 6.** Profiles across the deposits shown in Fig. 5. The profiles were obtained by integrating the height above the bed along the direction perpendicular to the respective profile, which can also be interpreted as volume per profile length. Solid lines refer to the axis-parallel alignment of the slope and dashed lines to the diagonal alignment.

the velocity may differ among parallel flow lines without any effect. However, transverse numerical diffusion results in an
exchange of momentum between different flow lines if the velocity is not parallel to any of the coordinate axes. This effect is obviously strong enough to suppress the formation of striations.

The occurrence of longitudinal striations is not unrealistic, and the modified Voellmy rheology may open a door towards understanding their origin. However, proceeding into this direction requires an extension of the model by transverse diffusion of momentum due to particle collisions in order to find out whether the tendency towards forming striations is strong enough
to overcome the diffusion of momentum.

The overall shape of the deposits is, however, not affected strongly by the orientation of the grid. This visual impression from Fig. 5 is confirmed by the profiles plotted in Fig. 6. Overall, these results suggest that there is no need to align the coordinate system to the principal orientation of the slope when simulating real-world scenarios since the local topography will override effects of the orientation.

**5.5 Fahrboeschung ratios**

Explaining the long runout of large rock avalanches was the main goal of developing the modified Voellmy rheology. The relation between runout and volume is often expressed in terms of the fahrboeschung ratio $\frac{H}{L}$, also called Heim's ratio, where





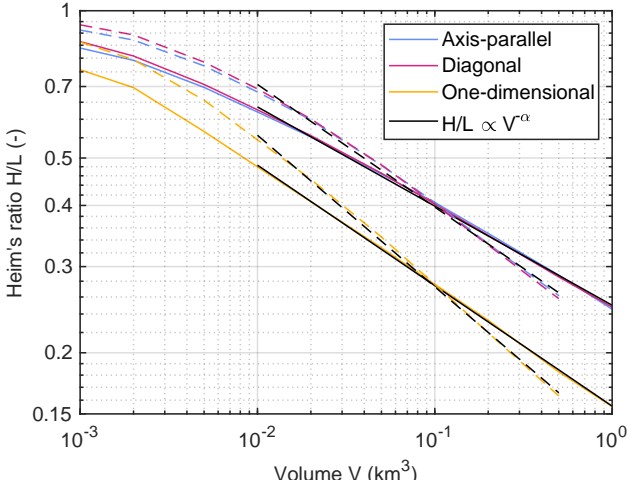

**Figure 7.** Heim's ratio as a function of the volume. Dashed lines refer to a total height of $H = 1900$ m as considered in Sects. 5.3 and 5.4. Solid lines were obtained taking into account the correlation between $H$ and $V$ observed by Legros (2002). Black lines represent power laws fitted for $V \geq 0.01$ km$^3$.

$H$ is the total height drop and $L$ the maximum horizontal runout length. Both properties are measured from the upper edge of the detached volume. Scheidegger (1973) found a power-law relation

$$\frac{H}{L} \propto V^{-\alpha} \tag{40}$$


with $\alpha = 0.16$, which was confirmed later by Legros (2002).

The dashed lines in Figure 7 show the relation obtained numerically with the volumes from the previous section extended to $V = 0.001$ km$^3$, $0.002$ km$^3$, $0.005$ km$^3$, $0.01$ km$^3$, ..., $0.5$ km$^3$. While the power-law relation between volume and Heim's ratio is reproduced qualitatively well for $V \geq 0.01$ km$^3$, the effect of volume on Heim's ratio is overestimated. Fitting power

laws yields $\alpha = 0.25$ for the 2D version and even $\alpha = 0.31$ for the 1D version.

This finding is in line with the results of Hergarten (2023c), who considered the same rheology in a lumped-mass model and found a strong influence of the absolute height drop $H$. While larger volumes tend to have a larger height drop in nature, the resulting increase in $L$ is weaker than the increase in $H$. So Heim's ratio increases with increasing $H$. In order to take this effect into account, the solid lines in Fig. 7 describe the same scenarios as before, but with $H \propto V^{0.09}$ as found empirically by

Legros (2002). Keeping $H = 1900$ m for $V = 0.1$ km$^3$ yields a range from $H = 1255$ m for $V = 0.001$ km$^3$ to $H = 2338$ m for $V = 1$ km$^3$. In contrast to the previous simulations, $V = 1$ km$^3$ is now possible without getting into conflict with the foot of the slope.

Including the correlation of $H$ and $V$ reduces the exponent $\alpha$ considerably. For the 2D version, $\alpha = 0.20$ is obtained, which is still higher than observed in nature. The residual deviation can at least partly be attributed to the spatial scaling assumed

here. For simplicity, the detached volume was assumed to scale isotropically (so with $V^{\frac{1}{3}}$ in each direction). As explained by Hergarten (2023c), however, the thickness should be the primary control on runout rather than $V$ itself. Larsen et al. (2010)



found the relation $V \propto A^{1.40}$ between volume and area, which means that increase in thickness is weaker than $V^{\frac{1}{3}}$ in reality. Since there are several dependencies beyond this scaling relation, the value $\alpha = 0.20$ seems to be good enough for a first test. Overall, these findings confirm the results obtained by Hergarten (2023c) for a lumped mass and suggest that the 2D scenario
without lateral confinement yields similar scaling properties as the simple 1D lumped-mass model.

## 6   Conclusions

In this study, a simple and lightweight numerical implementation of the modified Voellmy rheology proposed by Hergarten (2023c) was presented. The simplicity of the implementation arises from two features. First, a fully Cartesian description is used, where the thickness of the mobile layer is measured vertically instead of perpendicularly to the bed. This concept
harmonizes well with a simplified expression for the pressure at the bed. As a second feature, a simple upstream scheme is used for the advection terms. While upstream schemes typically suffer from numerical diffusion, it was shown that numerical diffusion is not a serious problem in combination with Voellmy-type rheologies.

The results obtained from the numerical tests confirm the findings of Hergarten (2023c) for the simple lumped-mass model. Furthermore, the modified Voellmy rheology may open doors towards understanding hummocky deposit morphologies and
longitudinal striations. In view of the simplicity of the numerical implementation, these preliminary results are very promising.

In turn, however, the purpose of the recent implementation must be kept in mind. The MinVoellmy software is lightweight, but minimalistic. It does not offer any user interface or methods for importing and exporting data. So it cannot be operated without programming some parts in MATLAB or Python, which limits its field to research and teaching. It is also not designed to compete with comprehensive models such as r.avaflow, which offer additional options such as multi-phase flow.

In particular, the recent implementation is not designed for operational hazard assessment. This restriction is not only owing to the short time span of development and limited testing, but mainly to the parameter values. The parameters $\mu$ and $\xi$ of the modified Voellmy rheology are similar to those of the conventional Voellmy rheology in their meaning, but their numerical values are not the same. So calibrations of other models, such as the extensively tested parameter values from the widely used model RAMMS, cannot be transferred to the modified rheology directly. Furthermore, knowledge about the additional
parameter $v_\mathrm{c}$ is still very limited.

*Code and data availability.*  All codes are available in a Zenodo repository at https://doi.org/10.5281/zenodo.7851614 (Hergarten, 2023b) and can be redistributed under the GNU General Public License. This repository also contains data obtained from the numerical simulations. Interested users are advised to download the most recent version of the MinVoellmy software from http://hergarten.at/minvoellmy (Hergarten, 2023a).

*Author contributions.*  S.H. developed the numerical scheme and the codes, performed the tests and wrote the paper.





*Competing interests.* The author declares that there is no conflict of interest.



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
