# Peer review of "Correspondence: Stefan Hergarten (stefan.hergarten@geologie.uni-freiburg.de)"

_EGUsphere, 2023_

## Author Response (AR1)

Dear Reviewers, dear Editor,

thank you for your comments, in particular to the second reviewer (Fabian Walter) who obviously read the paper thoroughly and spent much time! The points addressed in the two reports are discussed below, where changes to the manuscript are highlighted in bold letters. Line numbers refer to the version with highlighted changes.

In addition, I made a few more changes to the manuscript:

- I removed the discussion of the model proposed by Jop et al. (2005,2006) since I shortened it in my other paper about the modified Voellmy rheology and adjusted the discussion of the original Voellmy rheology (lines 34–45).

- I added a reference to a preprint about the new model AvaFrame com1DFA (Tonnel et al., 2023) (lines 60–64).

- I fixed some mistakes and typos.

Best regards,

Stefan Hergarten

**Reviewer 1**

*The paper introduces a numerical model based on a modified Voellmy friction model that is currently under review in a different manuscript. The mechanical concept follows Hergarten and Robl (2015).*

*From a scientific point of view, this work does not provide many novelties and innovations. The modified friction model is mentioned as the motivation for this work and seems to be its focal point. Therefore I would merge this manuscript with the other manuscript under review.*

*The model seems to follow the same concept as SHALTOP (Brunet et al. 2017) but comes to different conclusions and governing equations. SHALTOP also provides an exact solution of the Savage-Hutter model projected to a flat surface, while this work provides an approximation only. This should be discussed. What is the upside form the approximations done in this work? How do they compare?*

*Considering this is a new piece of code, I miss basic tests, e.g. comparisons with experiments, analytical solutions or existing software.*

The last statement is not correct. The approximations to the acceleration terms are quite different.

I guess that merging two manuscripts in journals with different foci is not thought to be a serious suggestion. Model description papers in GMD should not present scientific results (such as the runout scaling as a function of volume in the ESurf manuscript), while ESurf is not a good place for technical descriptions of software.

A far as I know, SHALTOP uses the concept developed by Bouchut and Westdickenberg (2004). Then measuring the thickness vertically and averaging the properties vertically instead of perpendicularly to the bed is the main difference (also compared to the original Savage-Hutter model). **I added this information to the abstract (lines 7–8),** although I hope that those readers who read more than the abstract will find it anyway. **I also added more discussion about the advantages of considering the vertical thickness (lines 108–122).** The numerical tests to illustrate that the necessary approximation to the pressure has not a big effect on the behavior were already present in the original manuscript (Sect. 5.1 and 5.2).

As a second aspect, it was also shown in the original manuscript that shock-preserving numerical schemes are not as important as usually assumed for a certain type of rheologies (if friction decreases with thickness). This is illustrated quite in detail, although probably unimportant for you because SHALTOP uses a shock-preserving scheme. Anyway, the manuscript is not about arguing against any established model, but rather for illustrating how simple the numerical treatment can be.

Beyond this, there might be people interested in modeling landslide runout who do not want to pick information about a model from multiple papers and to rely on the authors providing them with the source code, as it seems to be the case for SHALTOP.

I thought that the comparison with the analytical solution for a front (with and without the approximation) was such a test.

**Reviewer 2 (Fabian Walter)**

*This submission by Hergarten presents a new implementation of the shallow water approach to granular flow modeling under the assumption of friction that switches between a Coulomb and velocity-dependent parameterization. The novelty is the adaption of a Cartesian coordinate system, which makes the implementation more tractable but may cause numerical problems. With a redefined expression for the hydrostatic pressure, the author is able to produce simple granular flows that agree with conventional solutions in a bed-parallel coordinate system.*

*The manuscript is written well, and most parts are straightforward to follow. The findings are supported with test model runs and analytical arguments. My main criticism concerns a lacking discussion in the context of the Savage-Hutter approach to the shallow water approximation. In the detailed comments below, I elaborate on this point and in various other parts of the text I ask the author to add clarification. Overall, I enjoyed reading the manuscript and believe that with the additional clarifications it can be brought to publication quality.*

I tried to improve the discussion in comparison to the original Savage–Hutter model at some places (see detailed comments).

*I would like to stress that I am not a model developer and hence I had to go through the literature background to provide this review. This explains why I took longer to submit my feedback for which I apologize. In addition, some of my criticism may not be justified or my questions may be trivial to answer. Since a simplified model framework as presented here is of particular interest to the modeling novice like myself, my review should nevertheless be of use.*

Your arguments about the starting level make sense to me. The approach and the implementation are indeed simpler than in other models, so that there could be a chance to make the stuff accessible to researchers who are not so familiar with numerical modeling. On the other hand, however, a model description paper in GMD has to be concise. So it is still a tradeoff.

*Major comments*

*The reformulated pressure expression is aimed to circumvent the numerical singularity that arises for situations when bed and surface gradients are perpendicular. The author acknowledges that this situation is not representative for the Savage-Hutter approximation but of interest in view of numerical considerations. At this point I wonder if the defiance of the Savage-Hutter approximation can be dismissed in such an easy way: After all, their approach is not only a geometric argument on the bed-surface configuration. Instead, it is used to simplify the Navier-Stokes equations by neglecting terms with the help of scaling arguments. Can these scaling arguments be brought in agreement with the perpendicular bed/surface geometry? I suggest clarifying this point in the context of a discussion on the Savage-Hutter approximation.*

The key message should have been that it cannot be dismissed from a theoretical point of view, but can be practically. The problem is that the Savage–Hutter model was developed under mathematically well-constrained conditions, but practically often used at the edge of or even outside its theoretical field of applicability. **I added some more discussion about the Savage–Hutter model and what is different here (lines 108–122) in order to prepare the readers a bit more to the considerations coming later.** But anyway, the simplification of the Navier–Stokes equations by Savage and Hutter is in principle just deriving the approximation for the pressure, which is Sect. 2.1. Finally, however, the choice of the different approximation cannot be justified easily from a theoretical point of view. We can imagine that it makes no big difference because it is the same as the original Savage–Hutter model if the fluid surface is either parallel to the bed or horizontal. The numerical examples shown in Sect. 5.1 and 5.2 illustrate that it really makes not a big difference practically. The main problem is to assess the error arising from the approximation compared to the approximations already made by Savage and Hutter. Concerning the limitation of being only first order in thickness over radius of curvature of the Savage–Hutter model, the new Cartesian approach is presumably even better. However, a quantitative analysis of which approach is finally better is complicated, and I would prefer not to discuss the limitations of the Savage–Hutter model in a model description paper.

*Most equations were discussed and presented in a way that allows the reader to verify and understand them. However, I strongly suggest adding a sketch in which bed and surface are shown and angles are defined. This sketch should also define the signs of different quantities (e.g., angles and acceleration), which are crucial for the presented material.*

**I added three sketches – Fig. 1 with the geometry and the angles, Fig. 2 illustrating the effect of curvature and the limitation of the widely used version with the thickness normal to the bed, and Fig. 3 with the directions of the components of the acceleration.**

*Finally, the reader needs more information where certain equations come from or how assertions are justified. First, Lines 50-63 and Lines 80-84 include important statements without references. Second, the balance equations (6) and (7) are given without a reference. I was able to derive the former one from the form given in Savage and Hutter (1989), but the second one is not so straightforward. The reader should be presented with a clear source in the literature. Moreover, all assumptions that go into these equations should be stated (e.g., incompressibility?).*

**I added some more explanation about the balance equations and how they are related to those of the original shallow-water equations (lines 124–153).** However, I do not know whether these equations occur anywhere in the literature exactly in this form. Depending on coordinates, approximations, etc., the balance equations often look slightly different, but the structure is always similar. Concerning lines 50–63 (preprint), all specific information was already backed-up by references, except for the general statements about numerical diffusion and Lagrangian vs. Eulerian methods. Numerical diffusion has been widely studied, and Lagrangian vs. Eulerian methods are discussed in each introduction to continuum mechanics. From my point of view, adding randomly picked references would not make much sense. Finally, I am not sure what the important statements are in lines 80–84 (preprint).

*As another remark on these equations: please state if the del operators act on the product of $v_x$ and $h$ or on the first factor, only.*

**I adjusted the notation in Eqs. (6), (7), (23), and (24),** although it becomes more cumbersome and should be clear anyway.

*Specific comments*

*Equation 5: Here and elsewhere, are the bed and free surface $b(x, y)$ and $s(x, y)$ defined as level sets? I.e., $b(x, y) - z(x, y) = 0$? This should be stated.*

I am afraid that I missed your point. There is nothing like $z(x, y)$. It is just characterizing each point by $(x, y, z)$.

*Line 112: "where f is the absolute value" of what?*

Of the frictional deceleration. **I added an explanation (lines 144–145).**

*Equation (8): Is $p$ defined? Provide a reference for this equation. Perhaps trivial, but why is $a\dot{n} = 0$ a requirement?*

Unfortunately not, **but now it is (line 159).** However, Eq. (8) is the basis of all fluid dynamics, and I think it is not very useful to give references for such fundamental equations, given that even the explanation of the Navier-Stokes equations on Wikipedia starts from this point. **I added some explanation about the origin of the assumption** $a \cdot n = 0$ **(lines 160–164).**

*Equation (12): Is hydrostatic pressure assumed here? Or what is the motivation for this equation?*

Yes, basically the same as in the Savage–Hutter model. However, Eq. (12) does not require an assumption directly, but is derived from the previous equations. **I added some explanation (lines 160–164 and 170).**

*Line 134: See main comment above. Also, it may help referring to the trigonometric identity tan(theta)=-cot(theta-pi/2) to concisely pinpoint the singularity.*

Right, but I am not convinced that trigonometric identities are really helpful here. **I added the condition** $\phi = \beta - 90°$ **(line 180).**

*Figure 1: Here it would be extremely helpful to see a sketch for the geometry of the depicted situations (see main comment above). Similar, the asymmetry discussion (Lines 145ff) would benefit from a sketch depicting uphill-facing and downhill-facing fronts. Is "a" the acceleration in the bed-parallel direction? The dashed lines should be defined in the captions and not only in the main text. The description of the singularity makes sense, but I do not understand why only the dashed parts of the curve are shifted right under the coordinate transformation. Perhaps this can be rephrased.*

**The geometry (in particular the angles) should be clear now from the new Fig. 1.** However, I think that it should be clear in combination with the information $\phi < 0°$ what an uphill facing front is. Yes, $a$ is bed-parallel, as explained in the beginning of Sect. 2.2. **Anyway, I added it to the caption and also introduced dash-dotted lines for a better explanation. Furthermore, I rephrased the part about the dashed line segments (lines 189–190).**

*Line 159: Which "curves of the two models"?*

**I stated it explicitly now (line 210),** although I cannot imagine that it could have been unclear.

*Line 164: "the good properties of the equation": Which properties of which equations?*

Sorry, but it would not make sense to me to repeat all considerations of this section here.

*Line 169: This may sound like splitting hairs, but I would stick to acceleration and call it either uphill or downhill, ideally associating the two with a sign using a sketch (see major comments).*

It is not splitting hairs, but not correct. The direction of the frictional deceleration is related to that of the velocity and cannot be assigned to either uphill or downhill. **This will hopefully be clearer with the new Fig. 2.**

Lines 174-175: I suggest including a reference for "earth pressures" since this seems widely used in the soil mechanics literature.

Sorry, but what would be the use of such a reference? I am not so familiar with the soil mechanics literature and would even not be able to provide a textbook in which the explanation is better than the quite comprehensive explanation on the Wikipedia page.

Equation 21: Define $c$.

**I added a reference to Eq. (7) in order to remind the readers (line 229).**

Equation 22: Should min be max?

Indeed, **I fixed it (Eq. 22, lines 272, and Eq. 29).** Thanks!

Equation 23: The Euler scheme references I found include a $+$ rather than a $-$ sign between the two RHS terms.

The minus-sign arises from bringing the two terms to the right-hand side in Eq. (6). I feel that is would be too basic to be explained explicitly.

Equation 24 and the fowling equations: The primes are not derivatives, right? If so, I suggest specifying this.

Right, they are not derivatives. However, $hv'$ is already announced in the text before the equation as an "intermediate" value. I checked at https://en.wikipedia.org/wiki/Prime_(symbol) that the derivative is a quite specific usage of the prime symbol among many others. So I feel no need to clarify it explicitly.

Line 241: The referenced equation is a proportionality. How is the constant of proportionality determined?

Yes, the referenced equation ($v_c$ as a function of $h$) is a proportionality, and the parameter vc is $v_c(h)$ for $h = 1$ m. So vc defines the factor of proportionality. I am not sure whether or not this should be explained in more detail.

Line 255: Give a reference for the Courant-Friedrichs-Lewy criterion.

I am aware that some people want each word to be supported by a reference. However, even the respective Wikipedia page guides readers to the original work, although these papers would definitely not be helpful for novices.

Lines 259-260: Avoid 1-sentence paragraphs. Define the "rectangle".

**I moved the sentence to the description of the parameter $h_{\min}$ (lines 297–298)** because I do not find it useful to write more text just to avoid 1-sentence paragraphs. Anyway, I feel that a rectangle around a given set of points on a regular 2D grid needs no explicit definition.

Lines 267: Which "versions"?

**I specified it (lines 322–323),** although I thought it would be clear from the section heading.

Line 273: Rewrite "some waves".

**I rephrased it (line 328),** although I am not sure where the problem is whether it is better now.

Line 280: melts down $\rightarrow$ decreases

**Ok (line 335).**

*Line 288: "straight slope" is inappropriate since slope is a scalar value.*

No, slope is a geomorphic term here and not the slope of a curve. Convex, concave, and straight slopes are not uncommon terms in geomorphology and will probably not confuse the readers.

*Equation 37: Should the cosines be squared?*

**I added an additional step in order to show that one of the cosine factors vanishes,** although I am not sure whether it makes sense to get into such detail in each calculation.

*Lines 310ff: How is this supported? Is there a corresponding figure?*

The first and second results are immediately recognized in Fig. 3, which is still subject of the discussion here. For the second part (that widening ceases soon), there is no additional figure. Readers just have to believe that simulating the front over a distance of 10 km is sufficient.

*Figure 3: There seem to be several curves with the same color, please allow for better distinction. I could not distinguish lines with and without markers. For some of the lines in (b) there seems to be an overhang near $x = 0$. If intended, I suggest commenting.*

I am quite sure that the markers on the lines (and the kinks at these points can be recognized. Much larger markers would overlap for $\delta x = 10$ m, and using more colors would be confusing. **Concerning the overhang, I added some explanations about the coordinate system (lines 374–376),** but would leave to to the readers to understand the overhanging shape in detail.

*Line 331: Robustness with respect to what?*

**I removed the sentence (line 388)** since explaining it in detail would be too long.

*Line 332-333: "works well technically" should be qualified better or even quantified.*

**I replaced it with "remains stable" (line 390).**

*Figure 4: I suggest labeling the colors directly in the plot (e.g., with a legend or text boxes). The reader will appreciate this.*

**I added legends and fixed the wrong description in the caption.**

*Line 339: Delete one "necessary".*

**Fixed (line 396),** thanks!

*343: Why does the velocity remain constant at the kink? Because it is a single point in space?*

Ok, a bit sloppy. **I explained it in more detail (lines 401–403).**

*Figure 5: I suggest outlining the release areas rather by encircling it. Referring to them as "red lines" in the caption is confusing. Also, I strongly recommend pointing out the striations and hummocks discussed in the text directly in the figure.*

The red lines are the outlines. **I adjusted the caption in order to clarify this aspect.** However, I think the hummocky topography (variations in $h$) is easily recognized. Owing to the scale, recognizing the striations requires zooming into the figure. The problem is that both structures occur at many places in the figure, so that highlighting some of them explicitly would be rather confusing.

*Lines 368ff: It is not obvious to me why transverse diffusion is not allowed in Equation 7. After all, both horizontal components and gradients are present. Perhaps this is a well-known fact, but it would be interesting to have more information on this phenomenon.*

It is (i) because the advection part (left-hand side of Eq. 7) only transports momentum in direction of $v$ and not perpendicular to this direction and (ii) because the friction term at the right-hand side does not contain any derivatives of $v$. However, I think that your interest in this point is a bit specific since you started with some limited background concerning the theory, but dug quite deep into it now. Readers with a deep background in partial differential equations will recognize immediately that it is true, but "average" readers will probably not be so much interested in a detailed explanation since it refers to a minor result.

*Line 371: obviously $\rightarrow$ apparently*

**Ok (line 430).**

*Line 372: A reason or ideally a reference why the longitudinal striations are realistic should be given.*

**I added two references (line 431).**

*Line 373: into $\rightarrow$ in*

**Fixed (line 432),** thanks!

*Line 376: Rather than using "strongly" I suggest quantifying the grid orientation effect.*

I think that these examples are too specific, so that a quantification would be very limited. It should be clear that the systematic effect of the orientation is weaker than the other uncertainties, e.g., from the parameter values. Therefore, I think that "not affected strongly" describes it well.

*Line 381: I am surprised that this was the motivation for model development. After all, other models have reproduced long runouts. Perhaps a clarifying sentence would help.*

This may indeed be surprising for readers who know about the long-lasting discussion about the origin of the long runout. However, I feel that readers should consider the paper in which I introduced the modified rheology and that it would not make much sense to take up the discussion here.

*Figure 7: The different alphas should be labeled directly in the plot.*

**Ok.**

*Line 394: Which proportionality factor was used?*

It is implicitly defined in the next sentence. **Anyway, I rephrased it in order to make it clearer (lines 454–455).**

*Lines 396-397: Rewrite "without getting into conflict...".*

**Ok (line 456).**

*Line 419: What makes the other models more comprehensive?*

I gave an example of an additional option for r.avaflow, but I think it would not be useful to start reviewing the capabilities of other models in the conclusions section.

*Lines 420–421: Why would the time it takes to develop a model affect its use in hazard assessment?*

Perhaps many people would just not trust in my ability to develop a technically correct implementation within a few weeks or months.

*Lines 423–424: From my experience even when applying the same model to two different sites, parameter transferability is limited.*

I share the same experience. However, I feel that this paper is not the right occasion to question the applicability of RAMMS or other "established" models or the transferability of the respective parameter sets.

*Minor comments*

*Line 137: A singularity also exists if grad h grows without bounds. I suggest using other terminology.*

I would not agree. Consider, e.g., $f(x) = x^2$. Then also $f(x) \to \infty$ for $x \to \infty$, but we would not consider at a singularity.

*Fewer adjectives and adverbs reflecting subjectivity in argumentation should be used. Line 8: Delete "quite well". Line 60: Delete quite. Line 269: Delete very. Line 284: Rewrite works well. Line 347: delete quite. Line 425: Delete very.*

**I changed it to "fairly well" in line 10,** but I need something to make clear that it is not perfect here. **In lines 67 and 68, I removed "quite",** although I would still prefer a soft and somehow subjective statement here. **I also removed "very small" in line 324,** although it was intended to tell the readers that it is for sure small enough, without having to discuss it in detail. For "works well" in line 336, however, I have no idea what could be a better wording. **I removed "quite" in line 406 and "very" in line 475.**

*Line 37: Rewrite/correct "the lowest friction a low velocities".*

**I have rewritten the respective paragraph (lines 37–45),** so that this mistake has vanished.

*I may have missed it, but I could not find a plug-and-play matlab or python script on the code repository. It would be helpful if there was a setup that can simply be executed to reproduce the shown figures.*

The code repository contains the MATLAB scripts for all figures (named according to the figure number in the preprint). The scripts referring to the 2-D simulations (figure5.m, figure6.m, figure7.m) use pre-computed data in order not to recompute everything when modifying the figure. However, the scripts for computing these data are also included. Anyway, the code repository is still just made for this paper, and I prefer to provide later model versions and tutorials etc. via the "project homepage" at http://hergarten.at/minvoellmy.

---

## Author Response (AR2)

Dear Thomas Poulet, dear Fabian Walter,

thank you for your comments! The points are discussed below, where changes to the manuscript are highlighted in bold letters. Line numbers refer to the version with highlighted changes.

Best regards,

Stefan Hergarten

**Reviewer 2 (Fabian Walter)**

*The manuscript version by Stefan Hergarten has undergone improvements in clarity and explanations to bring it up to publication quality. At this point, my main remark concerns the extent at which the reader should be presented with background information to understand the paper's findings. My point of view is skewed, since I am not a modeling expert. Nevertheless, I restate those remarks, which the author finds are unnecessary to address below and leave it up to the editor to make a decision. In addition, I list minor comments and corrections.*

*I want to thank Stefan Hergarten for this accessible contribution and his explanations in the rebuttal letter. I learned a lot!*

Nice to hear! I felt a bit bad since you obviously spent much more time than reviewers typically do, but the time was not entirely lost then.

*COMMENTS*

*From the rebuttal letter, I understand that the author does not share my point of view that all stated information including equations that are not straightforward to derive have to be referenced. I still believe that this is essential for technical writing and do not share the view that information can be assumed known if it appears on Wikipedia articles or in the form of similar but not identical equations in the scientific literature. Adhering to this practice makes information tractable and avoids propagation of incorrect assumptions and findings.*

I learned that many authors adopt equations and even results with references, but without understanding the context. I honestly believe that this practice contributes more to the propagation of incorrect assumptions and findings than writing fundamental equations without a reference. Since all equations are developed step by step (perhaps sometimes a bit too fast), I guess that the problem is still the shallow-water equations (Eqs. 6 and 7) and the simplest form of the Navier-Stokes equations for an inviscid fluid (Eq. 8). **I added some more details about what can also be found in the book by Vreugdenhil and what is new here (lines 124–133).** Concerning the acceleration term (starting from Eq. 8), it was already stated in line 134 that it is the same as assumed by Savage and Hutter (1989). I simply do not want references to papers which also used the Navier-Stokes equations for justifying such fundamental equations.

*Abstract: I find the last sentence unnecessary at this point of the manuscript.*

I also do, but I remember that other people from your institution almost forced me to state it each occasion that only RAMMS should be used for operational hazard assessment. **Anyway, I am happy to remove this sentence from the abstract (lines 11-12).**

*Lines 59 and 60: A sentence stating why a particle-following coordinate system avoids numerical diffusion would be helpful.*

I would already have written such a sentence if it was easy. People who are familiar with the numerics of advection problems already know this, but it takes a full lesson to explain it to students. If a reader wants to go deeper into this topic, a search for the keywords "Lagrange numerical diffusion" already yields good documents beyond the two references to the available Lagrangian models.

*Line 89: GIS should be spelled out, especially since this acronym is never used again.*

**Ok (line 79),** although I would not expect that using the acronym without spelling it out would be a problem for any reader.

*Line 126: occur → appear*

**Ok (line 116),** although I am not completely sure.

*Line 190: Not sure what is meant by "come into play".*

**I rephrased it (lines 179–180).**

*Line 210: Versions of what? Perhaps better "expressions of the acceleration a"?*

**Ok (line 200),** indeed better.

*If I understand correctly, Figure 6 illustrates Equation 39 as well as an equivalent expression for the original pressure. It does not seem straightforward to derive the latter, so seeing a few mathematical steps or comments would be helpful.*

Indeed not straightforward, but requires repeating the steps of Eqs. (37) to (39). **I added one intermediate step and the final result (lines 361–366),** although I find adding more and more information about not very relevant steps rather distracting than helpful.

*Line 369: "be the front" → typo*

**Fixed (line 355),** thanks!

*Figure 7: I suggest stating the reduced friction coefficient in the figure or its caption.*

**Ok, I added it to the caption.**

*Line 408: "is piles up" → typo*

**Fixed (line 396),** thanks!

*Figure 8: I have to admit that I still cannot identify the hummocks and striations. To me, neither of the two deposits for $V = 0.5$ km$^3$ seem more or less hummocky or striated. As a result the discussion is enigmatic to me.*

Maybe you are looking for features different from those I think of. **I added two close-ups for illustrations, hoping that it becomes clearer what I think of.**

*Concerning the software repository, not all the matlab codes compile (see errors below). It would be helpful if the numbers in the matlab script names (Figure\*.m) correspond to the figure numbers. Not all figures are represented in the matlab scripts.*
*figure5.m: Unable to perform assignment because the size of the left side is 1-by-801 and the size of the right side is 1-by-2. Error in figure5 (line 22) s(i,:) = size(im{i});*
*figure7.m: Error using load Unable to find file or directory 'paral0001.mat'. Error in figure7 (line 17) load(filename)*

I just did not want to create a new version of the repository for each round of reviews. So the codes were still numbered according to the figures in the preprint. **I adjusted the numbers accordingly.** Figures 1 and 3 are, however, not code-generated. **In order to fix the problems you noticed, I added comments to make clear that the data files to be loaded must be either recomputed or unzipped from the file data.zip.**